

# Differences in cardiovascular disease mortality between northern and southern China under exposure to different temperatures: a systematic review

Guangyu Zhai[1], Ziqing Jiang[1] and Wenjuan Zhou[2]

[1] School of Economics and Management, Lanzhou University of Technology, Lanzhou, Gansu, China
[2] Network Center, Gansu Provincial Hospital, Lanzhou, Gansu, China

## ABSTRACT

**Background**. Due to differences in climate and other environmental factors, exposure to different temperatures in China has different effects on the relative risk (RR) of cardiovascular disease (CVD) mortality. It is therefore important to compare the effects of exposure to different temperatures on CVD mortality in different regions of China.
**Methods**. To compare these effects, we performed a meta-analysis of 21 studies identified by a search of the Web of Science and China National Knowledge Infrastructure databases from January 1, 2014 to January 1, 2024. We performed the Cochran $Q$ test and $I^2$ statistics test to evaluate heterogeneity and Egger's test to evaluate publication bias.
**Results**. The pooled estimated size of the relationship between exposure to different temperatures and CVD mortality was 1.60 (95% confidence interval [CI]: [1.42–1.80]) for the extreme cold, 1.17 (95% CI [1.10–1.25]) for the extreme heat, and 1.16 (95% CI [1.10–1.24]) for extremely high diurnal temperature range (DTR). The Egger's test showed potential publication bias in studies analyzing both the extreme cold and the extreme heat.
**Discussion**. Extreme cold, extreme heat, and extremely high DTR are associated with an increase in CVD mortality in China, with extreme cold having the most significant effect. Residents of northern regions are more susceptible to high temperatures, while residents of southern regions are more sensitive to low temperatures.

## INTRODUCTION

Climate change is both an important public health issue faced by many countries and a significant factor affecting individual human health around the world (*Kouis et al., 2019*). The best adaptation strategies to climate change require a thorough and in-depth understanding of the nature of the effects of temperature on human health. Cardiovascular disease (CVD), a term that encompasses diseases that threaten the function of the human heart and blood vessels, is not only one of the most serious chronic health problems worldwide, but is also the leading cause of mortality (*Tang et al., 2022*). According to estimates from the Global Burden of Disease (GBD) study, the global burden of CVD has

Corresponding author
Ziqing Jiang, 1531661700@qq.com

steadily increased over the past 20 years, leading to 18.6 million deaths in 2019 (*Roth et al., 2020*). More than 75% of CVD-related deaths occur in low- and middle-income countries, including China, which has a very high CVD burden and CVD mortality (*Gheorghe et al., 2018*). The pathophysiological nexus between climate change and cardiovascular health has been rigorously substantiated by scientific research. Specifically, *De Vita et al. (2024)* contend that climate change, mediated by extreme temperatures, can disrupt multiple regulatory pathways within the cardiovascular system, exacerbating the vulnerability of individuals with pre-existing cardiovascular conditions, and thereby contributing to a notable rise in cardiovascular mortality. *Giorgini et al. (2017)* argue that extreme high and low temperatures inflict significant damage on the human cardiovascular system through a complex array of physiological processes, including disruption of hydration balance, modulation of sympathetic nervous excitability, activation of the renin-angiotensin system, and induction of inflammatory responses, all of which act directly or indirectly to compromise cardiovascular health. Therefore, the widespread concern over the influence of temperature on cardiovascular disease mortality, coupled with a rigorous exploration of the specific impacts of distinct temperature exposure types, is of paramount significance in devising effective prevention strategies and adaptation measures aimed at safeguarding human health.

There are three types of temperature exposure: heat (thermal), including high temperatures, extreme heat, and heat waves; cold, including low temperatures and extreme cold; and temperature variation, including the temperature range between day and night and the temperature between two adjacent days (TCN) (*Wu et al., 2023*). Previous studies have shown that average temperature is the best predictor of mortality (*Anderson & Bell, 2009*). As average daily temperature serves as an exposure indicator representing exposure during the entire day and night, it can therefore be used as a temperature indicator to measure extreme cold and extreme heat (*Guo et al., 2014*). As an important meteorological index related to global climate change, diurnal temperature range (DTR), which is defined as the difference between the daily maximum temperature and the daily minimum temperature, can be used as an indicator to estimate the impact of temperature change on human health (*Kim et al., 2016*).

Many previous studies have shown that exposure to specific temperatures has a significant effect on CVD mortality in a given city in China. A study in Ganzhou showed a J-shaped relationship between average temperature and CVD mortality, with very low temperatures significantly increasing the relative risk (RR) of CVD death (*Zhang et al., 2021*). Another study in Qingdao reported that the RR of cardiovascular death at $-4.8\ ^\circ\mathrm{C}$ was 2.05 (95% confidence interval [CI]: 1.55–2.69) (*Zhai et al., 2022*). Other studies have revealed a nonlinear association between DTR and CVD mortality in Hulunbuir, and that very high DTR has a significant short-term adverse effect in terms of increasing CVD mortality (*Kai et al., 2023*). Most studies on the effect of temperature on CVD mortality have been limited to a single city, so generalization of their results is limited. The scope of the few studies that have conducted meta-analyses of the relationship between ambient temperature and CVD mortality was the entire Earth, so their results are too general for application to a specific region.

As China is a vast country with great geographical and climatic differences between the North and the South, the research findings on the impact of CVD in different regions can differ significantly. Therefore, further studies are urgently needed to fully understand the impact of temperature exposure on CVD mortality in all of China.

To help fill this research gap, we conducted a meta-analysis to leverage and integrate the existing literature reporting the results of studies that examined the effects of exposure to different temperatures and included consideration of the extreme cold, the heat (thermal) effect, and the DTR effect on CVD mortality in China. We stratified the analysis by the North and South to specify the effects in different regions. Our findings provide a better understanding of the impact of exposure to different temperatures on CVD mortality in China and a new theoretical basis and scientific reference for formulating different public health policies according to region.

## MATERIALS & METHODS

### Survey methodology

We conducted this systematic review and meta-analysis in accordance with the Systematic Review and Preferred Reporting Project for Meta-Analysis (PRISMA) guidelines (*Page et al., 2021*). The recommendations are based on an online accessible registry evaluation scheme (PROSPERO CRD42024550796) and the Meta-analysis of Observational Studies in the Epidemiology (MOOSE) checklist (*Stroup et al., 2000*).

We searched two databases, the Web of Science and China National Knowledge Infrastructure (CNKI) databases, using a combination of search terms related to temperature, CVD, and mortality in three stages. In the first stage, we searched using two terms, "temperature" and "cardiovascular disease". In the second stage, we defined the research area as "China", and limited the publication date to January 1, 2014 to January 1, 2024. In the third stage, we reviewed the article titles and abstracts to filter out research unrelated to the research question. We aimed to identify studies that explored the effect of different measures of temperature exposure on CVD mortality that met three criteria: (1) daily mean temperature or DTR must be used as an indicator of temperature exposure, (2) the research must have been conducted in China, and (3) studies must explain the association between temperature exposure and cardiovascular mortality.

### Data extraction

Every study was examined to extract the subsequent data: author, location, study duration, metrics of temperature exposure, lag days, RR of extreme heat and extreme cold, extremely high DTR, 95% confidence intervals, as well as subgroups categorized by region.

### Statistical analysis

The analytical process consisted of three pivotal phases, including: (1) the calculation of aggregate estimates for diverse sites in China, (2) conducting subgroup analysis according to region, and (3) performing deviation test and sensitivity analysis. In the first phase, we performed the meta-analysis approach to calculate combined estimate of the RR for distinct locations, culminating in the production of a forest plot for visual illustration of

these estimates. We also assessed heterogeneity in the relationship between exposure to different temperatures and the RR of death from CVD. We assessed heterogeneity using the Cochran Q, considering $P$ values less than 0.05 as an indication of significance, and we used $I^2$ statistics to classify heterogeneity as low ($\leq$25%), medium (26%–74%), or high ($\geq$75%). Specifically, the standard error is estimated by collecting the RR values and their lower and upper limits of the 95% confidence intervals from each study. Following this, an appropriate model is selected. If significant heterogeneity exists, a random effects model is chosen; otherwise, a fixed effects model is utilized. Based on the selected model, summary RR values are calculated separately for conditions of extreme cold, extreme heat, and extremely high DTR. Finally, we performed Egger's test to assess potential publication bias. In the second phase, we conducted subgroup analysis of the extreme cold, the extreme heat, and extremely high DTR. To delve deeply into and clarify the potential factors that give rise to this observed significant heterogeneity, we conducted subgroup analyses stratified by region (North and South), and presented the results in the form of forest plots. In the third phase, we employed a sensitivity analysis approach, which systematically removed one study from the analysis in succession to test the impact of individual estimates on the pooled RR and the robustness of the examination results. We performed all statistical analyses using the Meta package in R (v.4.1.2).

## Assessment of evidence

The quality and strength of the evidence regarding the associations between exposure to extreme cold, extreme heat, and extremely high DTR with CVD mortality were meticulously and independently assessed. Detailed information on quality assessment is provided in Table S1.

## RESULTS

Figure 1 illustrates the steps involved in the inclusion and exclusion of studies, resulting in the retrieval of 20,071 articles related to temperature and CVD from reputable databases such as Web of Science and CNKI. After adding the subject term "mortality", 2,238 articles remained in our sample. After the addition of several geographical terms that indicated that China was the study area, 613 articles remained in our sample. After the final review of the titles, abstracts, and full text, we included 21 articles, of which 12 reported the extreme cold, 11 reported the extreme heat, and eight reported DTR, in our final sample (*Bai et al., 2014*; *Ding et al., 2015*; *Hu et al., 2019*; *Kai et al., 2023*; *Li et al., 2023*; *Liu et al., 2020*; *Liu, 2022*; *Luo et al., 2013*; *Tang et al., 2018*; *Wang et al., 2014*; *Wang et al., 2021*; *Xia, 2023*; *Xia et al., 2023*; *Xiao et al., 2021*; *Xu et al., 2022*; *Yang et al., 2015a*; *Yang et al., 2015b*; *Zhai et al., 2022*; *Zhang et al., 2018a*; *Zhang et al., 2018b*; *Zhang et al., 2021*).

Figures 2 and 3 present distinct reports on the combined effect of the relationship between daily average temperature and the RR of CVD mortality, segmented into extreme cold and extreme heat conditions. Under extreme cold conditions, analysis utilizing the random effects model reveals that the RR of CVD mortality escalates to 1.60 (95% CI [1.42–1.80]). Under extreme heat conditions, while the RR of CVD mortality also demonstrates an increasing trend, reaching 1.17 (95% CI [1.10–1.25]), the magnitude of this increase is

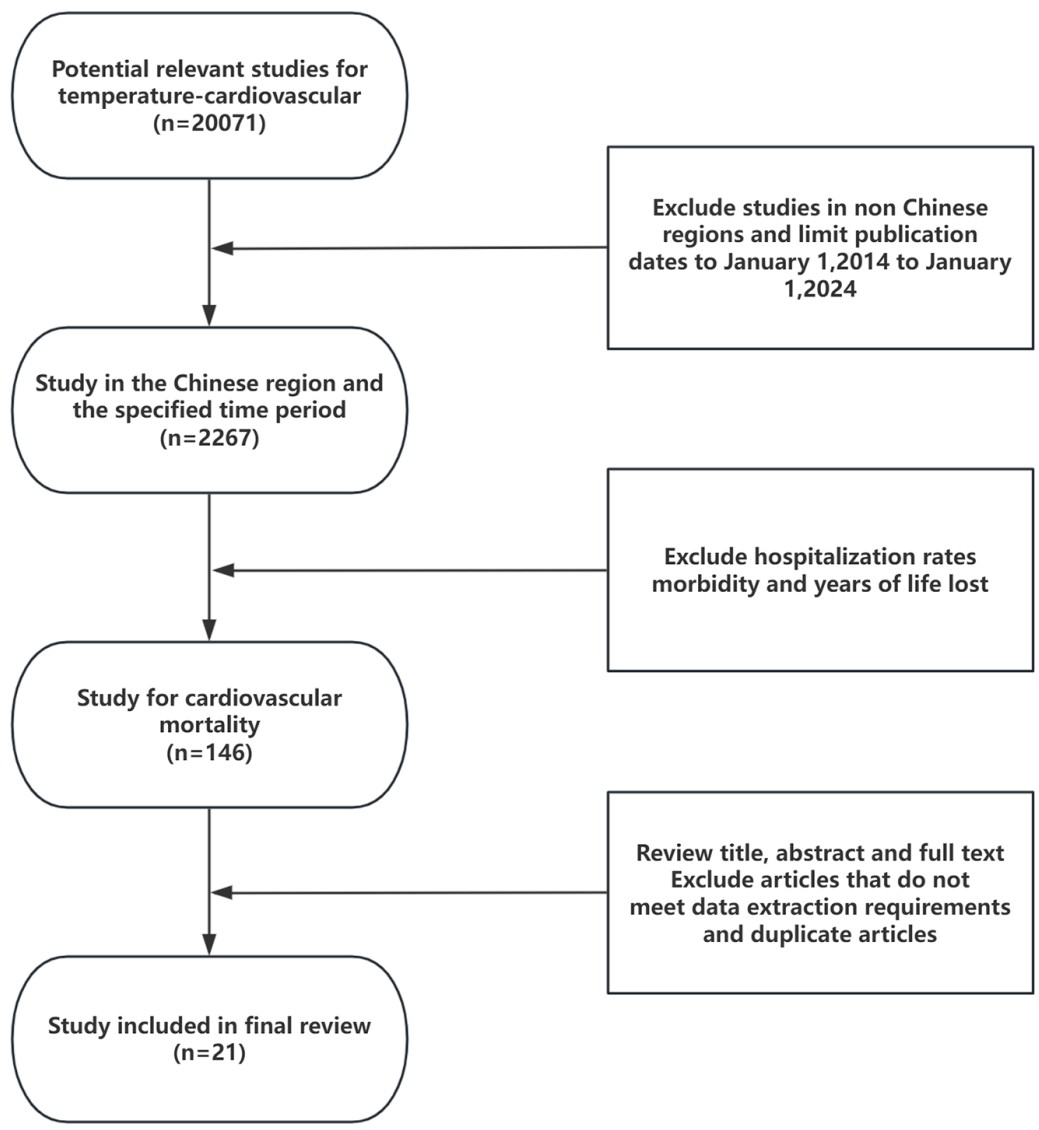

**Figure 1** Flow diagram of inclusion and exclusion of studies in the final sample.

comparatively smaller. It is noteworthy that the percentage of heterogeneity among studies, which refers to the variation in the results obtained from different research, is considerably high. Specifically, the $I^2$ value is 94% under extreme cold conditions and 78% under extreme heat conditions, both of which reach statistical significance at the $P < 0.05$ level. Therefore, we have chosen to adopt the results from the random effects model, as it takes into account the variability and potential unobserved factors among studies, enabling a more accurate reflection of the heterogeneity present across different research endeavors. Figure 4 shows that for extremely high DTR, our utilization of the common effects model revealed a relative risk (RR) of 1.16 (95% CI [1.10–1.24]) for cardiovascular disease (CVD)

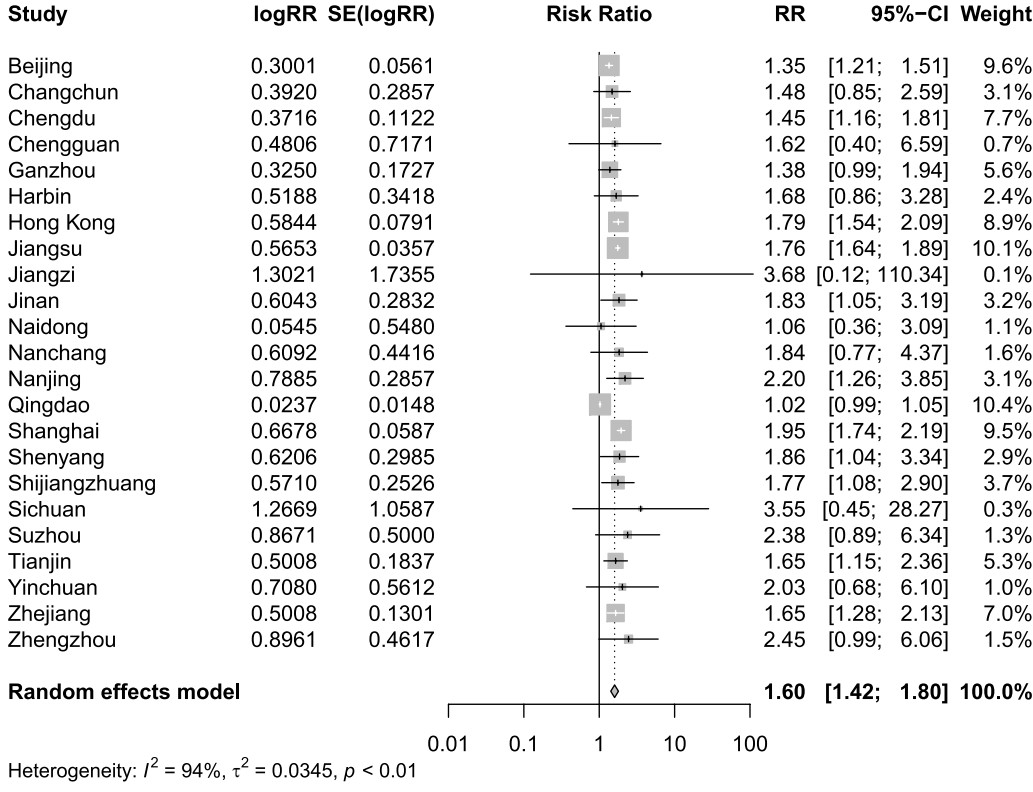

**Figure 2** Meta-analysis of extreme cold on relative risk of cardiovascular disease mortality.

mortality. Notably, the low heterogeneity among the included studies, evidenced by an $I^2$ value of 0% ($P > 0.05$), justifies our choice of the common effects model results.

Finding significant heterogeneity in the meta-analysis of all the included studies, we performed subgroup analyses for region, in an effort to delve into and elucidate the underlying factors contributing to this observed variability. Figure 5 shows that the combined effect of the North subgroup on CVD mortality under the extreme cold was 1.48 (95% CI [1.22–1.79]) and that of the South subgroup was 1.76 (95% CI [1.64–1.89]). Figure 6 shows that the combined effect of the North subgroup on CVD mortality under the extreme heat was 1.19 (95% CI [1.09–1.31]) and that of the South subgroup was 1.15 (95% CI [1.05–1.26]). Figure 7 shows that the combined effect of the South subgroup on CVD mortality under extremely high DTR was 1.16 (95% CI [1.06–1.26]) and that of the North subgroup was 1.17 (95% CI [1.08–1.27]). Heterogeneity was highly reduced when the analyses were stratified by region subgroup, with an $I^2$ of 82% for the North and 0% for the South under the extreme cold, 80% for the North and 67% for the South under the extreme heat, and 0% for both the South and North under extremely high DTR.

To assess the influence of sequentially eliminating studies from the analytical process on the comprehensive outcomes, we conducted a rigorous sensitivity analysis encompassing the incorporated studies. Figures 8, 9 and 10 show the results of the sensitivity analysis of the extreme cold, extreme heat, and extremely high DTR, respectively. In all three

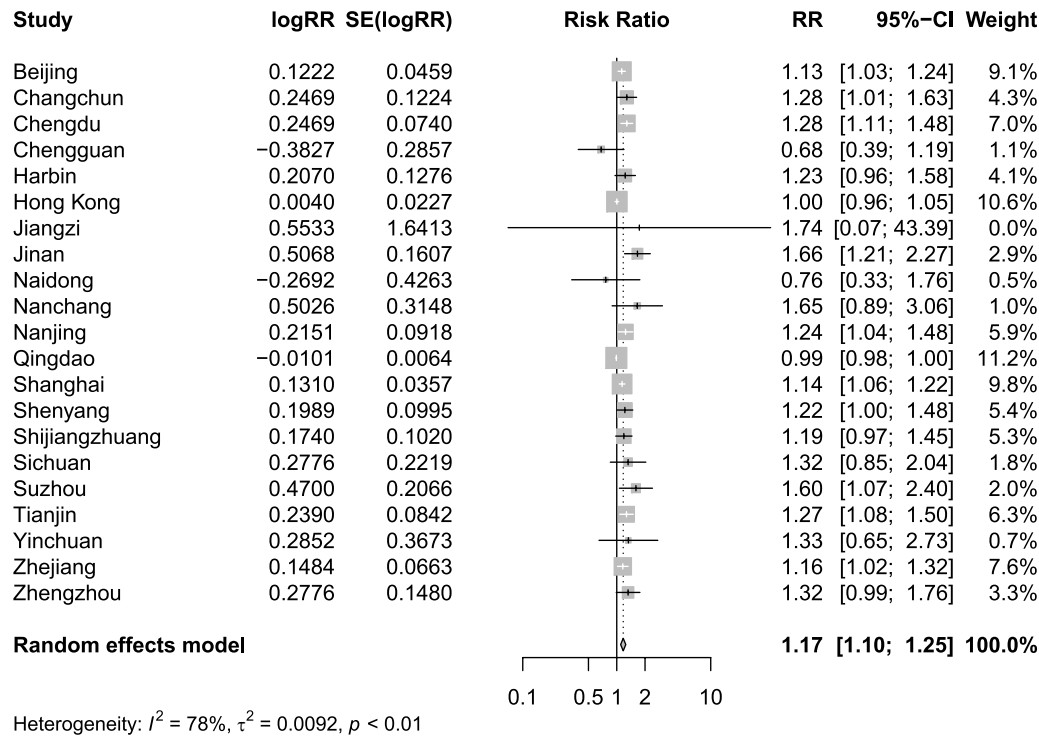

**Figure 3** Meta-analysis of extreme heat on relative risk of cardiovascular disease mortality.

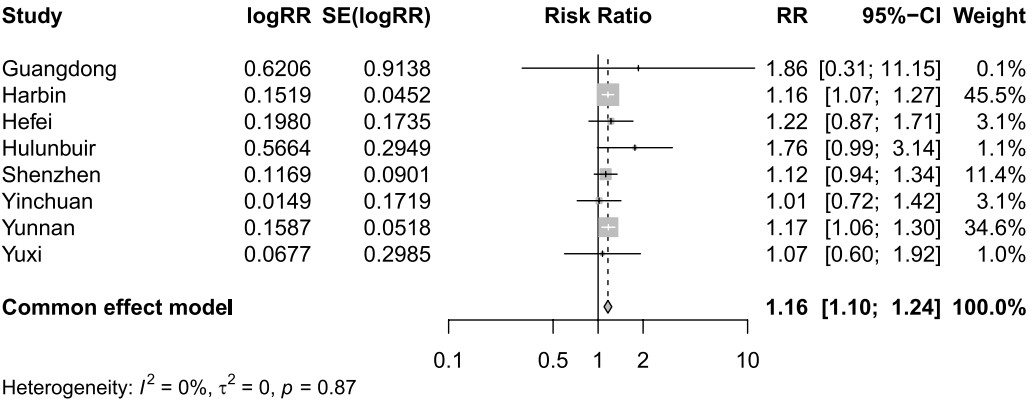

**Figure 4** Meta-analysis of extremely high diurnal temperature range on relative risk of cardiovascular disease mortality.

cases, there were no significant differences between these results and the result obtained before exclusion, which was 1.60 (1.42–1.80) for the extreme cold, 1.17 (1.10–1.25) for the extreme heat, and 1.16 (1.10–1.24) for the extremely high DTR. This indicated that the sensitivity analysis was good and the results were relatively stable. We performed Egger's

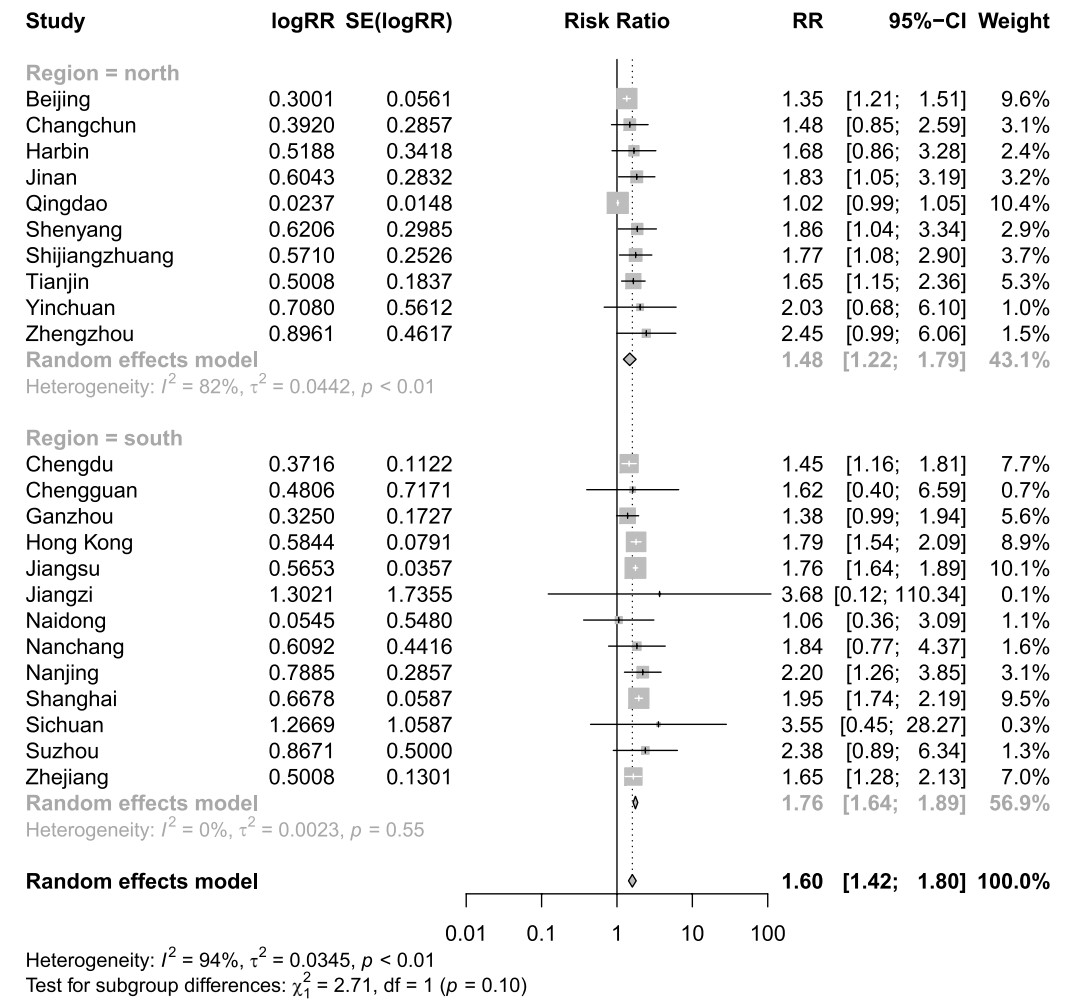

**Figure 5** Meta-analysis of extreme cold and relative risk of cardiovascular mortality by region.

test to evaluate the publication bias in the literature. The $P$ values for both the extreme cold ($P = 0.0015$) and the extreme heat ($P < 0.0001$) were less than 0.05, indicating the existence of publication bias between studies examining the extreme cold and the extreme heat.

## DISCUSSION

To the best of our knowledge, this was the first meta-analysis study to compare the impact of exposure to different temperatures in China on CVD mortality. We investigated the effects of daily average temperature and the diurnal temperature range on the RR of CVD mortality in most provinces and cities in China to identify the most significant factors in CVD mortality. We also measured the publication bias in the literature reporting on the extreme cold and extreme heat. Because we obtained data from published studies, our results may reflect publication bias (*Yin et al., 2020*). Studies reporting on intervention

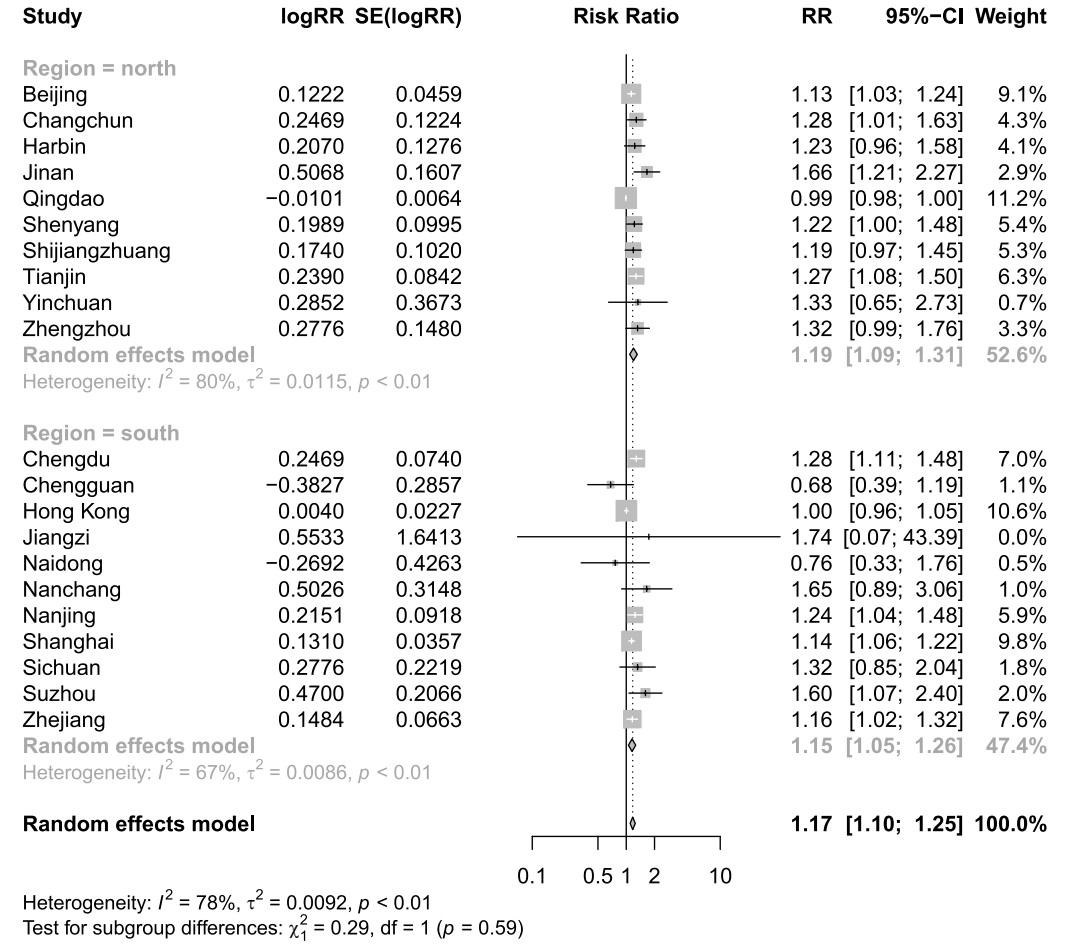

**Figure 6** Meta-analysis of extreme heat and relative risk of cardiovascular disease mortality by region.

measures that yield results significantly different from traditional treatments (positive results) are more likely to be published than studies reporting on intervention measures that yield results not significantly different from traditional treatments (negative results), the latter of which are often overlooked. The direction and strength of the research results determine the likelihood of publication. As a result, not all studies conducted are published, which can lead to publication bias.

Many studies have found that both extremely high and extremely low temperatures can increase the RR of CVD mortality (*Wang et al., 2015*). Our study confirmed the findings of previous studies. As discussed in the published literature, exposure to low temperatures may lead to increased blood pressure, blood viscosity, and plasma cholesterol, which may cause arterial thrombosis and increase the RR of CVD mortality through increasing blood concentration (*Huynen et al., 2001*; *Shiue & Shiue, 2014*). Exposure to extremely high temperatures, however, may cause vasodilation and increased sweating, which may lead to increased blood concentration, blood viscosity, and cholesterol levels. By analyzing eight cities in northern and southern China, we also found that an increased RR of CVD

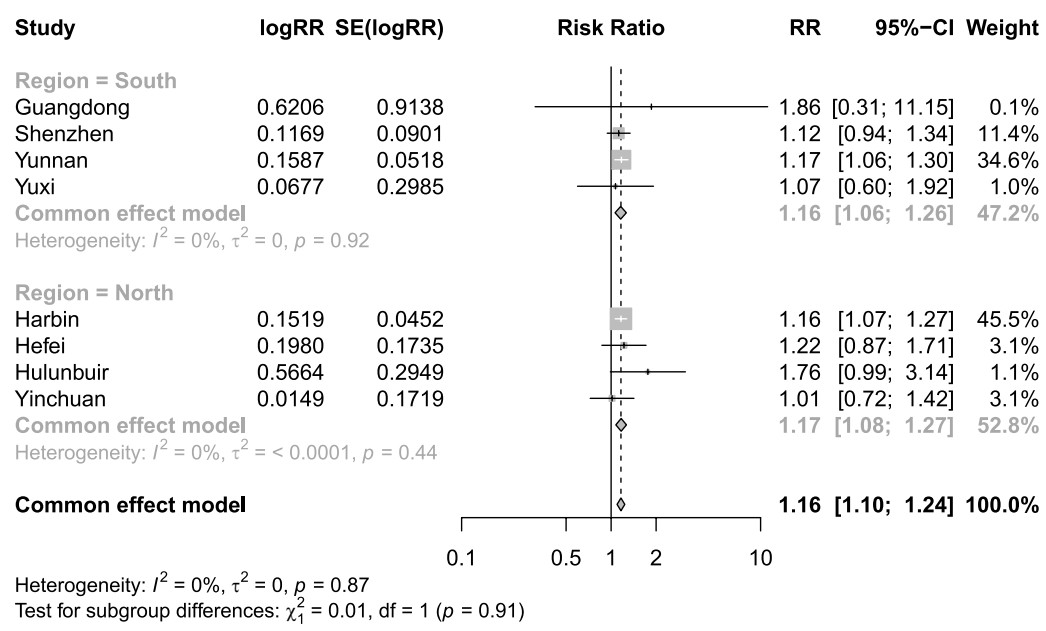

**Figure 7** Meta-analysis of extremely high diurnal temperature range and relative risk of cardiovascular mortality by region.

mortality in China was associated with an extremely high DTR. Many domestic and international studies have found that large temperature differences can increase the RR of CVD mortality, with a J-shaped relationship existing between DTR and CVD mortality (*Ding et al., 2015*; *Vutcovici, Goldberg & Valois, 2014*). For example, *Zhou et al. (2014)* found a linear relationship between DTR and mortality rate such that the mortality rate increased with an increase in DTR. In a study in Guangzhou, *Yang et al. (2013)* study found that for every 1 °C increase in DTR, CVD mortality increased by 0.75% (*Ding et al., 2015*). *Ding et al. (2016)* also confirmed that CVD death occurs more frequently on extremely high DTR days. The potential physiological mechanisms underlying the impact of DTR on CVD mortality are not fully understood. Studies have shown that sudden temperature changes may release inflammatory mediators associated with mast cells, thereby increasing cardiopulmonary load and inducing cardiopulmonary events (*Imai et al., 1998*; *Keatinge et al., 1986*). An increase in DTR may also increase the RR of CVD by increasing blood pressure, oxygen uptake, heart rate, and cardiac workload and may lead to arrhythmia, which may be explained by the mechanism underlying the relationship between CVD death and DTR (*Lim et al., 2013*).

By analyzing and comparing the effects of exposure to different temperatures on CVD, we found that extreme cold has a much greater impact on CVD-related mortality than either extreme heat or extremely high DTR. Our findings agree with those of *Huang, Wang & Yu (2014)*, who suggested that low temperatures have a strong cumulative effect on the RR of CVD mortality. They also agree with those of *Yang et al. (2015b)* who found that the extreme cold accounted for over 90% of the temperature-related CVD mortality

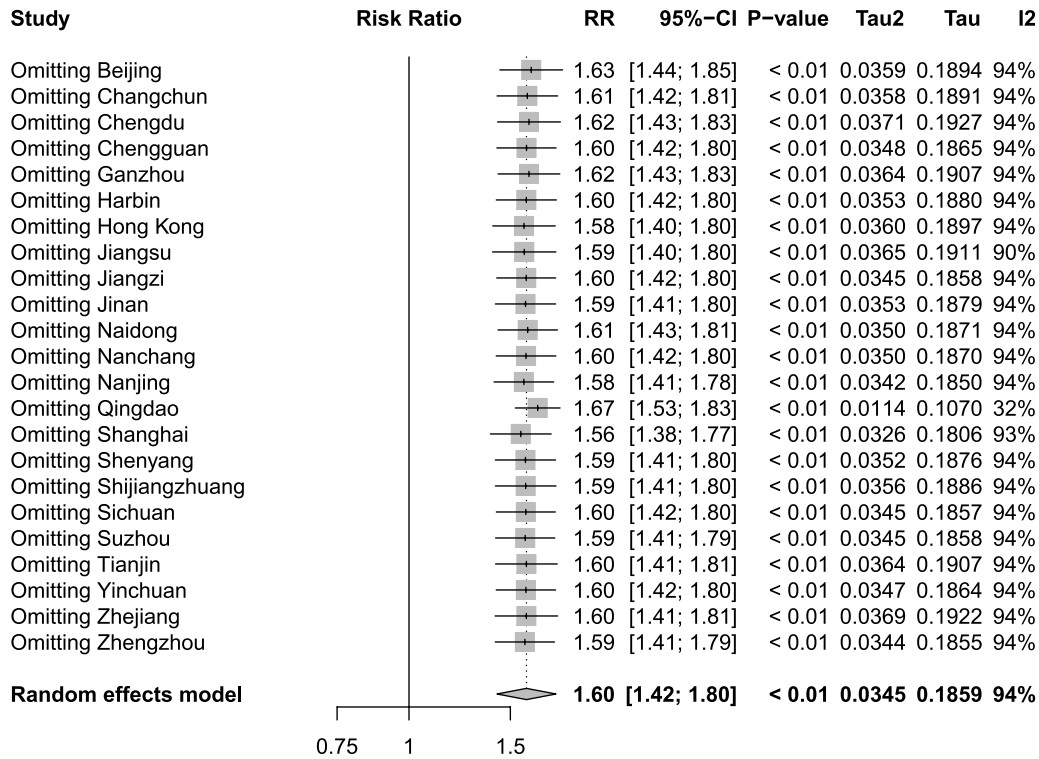

**Figure 8** Sensitivity analysis of extreme cold.

rate and that low temperature plays an important role in the winter excess mortality rate due to CVD. These findings can be reasonably explained by the fact that people reduce their frequency of leaving the home under extremely high temperature conditions and increase their frequency of using air conditioning to lower the indoor temperature, thereby protecting them from the effects of high temperatures (*Zhai, Zhang & Chai, 2021*). They may also take timely measures based on changes in the outdoor temperature to avoid the health effects caused by significant temperature changes (*Kai et al., 2023*).

Existing epidemiological evidence suggests that latitude may alter the relationship between temperature and mortality rate (*Xiao et al., 2015*). Based on the boundary between the Qin Mountains and the Huai River near 33° north latitude, we divided China into the North and the South. We found that under the extreme heat, the RR of CVD mortality in the North is higher than that in the South; in contrast, under the extreme cold, the RR of CVD mortality in the South is higher than that in the North. This finding indicates that residents of northern regions are more susceptible to the effects of high temperatures, while residents of southern regions are more susceptible to the effects of low temperatures. One possible reason is that heating equipment is widely used in the cold season in northern regions, typically from November to mid-March, making the indoor temperature comfortable and thereby reducing the likelihood of exposure to ambient temperatures (*Zhang et al., 2018b*). The higher extreme cold in southern regions may be evidence of adaptation to

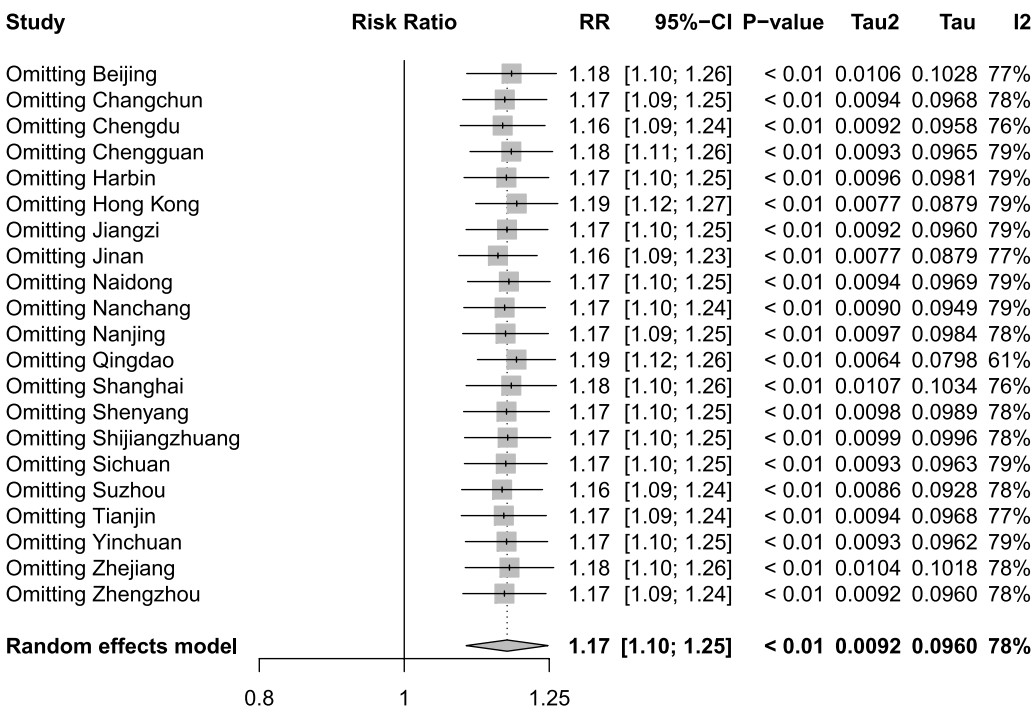

**Figure 9  Sensitivity analysis of extreme heat.**

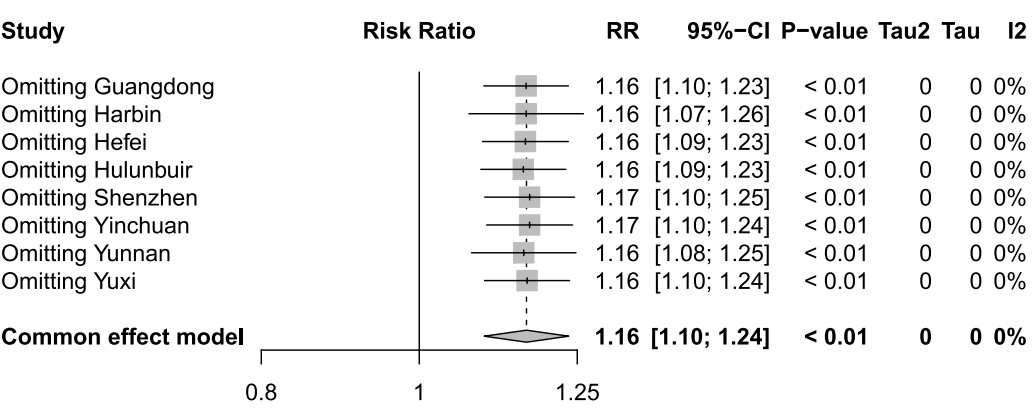

**Figure 10  Sensitivity analysis of extremely high diurnal temperature range.**

the environment. Although winter temperatures in the South are higher than those in the North, the lack of centralized heating systems in the South makes its residents more susceptible to cold weather, resulting in a higher RR of CVD mortality (*Xie et al., 2013*). This conjecture is consistent with the results of a study conducted in China showing that central heating may alter the temperature–mortality rate correlation and its lagged structure (*Chen et al., 2018*). Another possible reason is that thermal adaptability differs between residents of northern and southern cities. For example, *Lee et al. (2012)* found

that residents living in the tropics have the cardiovascular advantage of heat adaptation, during which plasma volume and stroke volume increase, maintaining cardiac output at lower heart rates. *Li et al. (2023)* hypothesized that northern cities are vulnerable to heat adaptation, which may lead to an increase in the impact of high temperatures on CVD mortality. However, they found no significant difference in CVD mortality between the North and South at extremely high DTRs. This finding is consistent with a previous study that reported that the results of analysis of the association between DTR and mortality are unlikely to be materially altered by geographic, climatic, or demographic characteristics, nor by publication bias or model size (*Zhou et al., 2014*).

The study shows the intricate interplay between geographic location, climate conditions, and cardiovascular health. Climate change has the potential to exacerbate the frequency and intensity of extreme weather events, posing increasingly serious challenges to public health (*Yang et al., 2017*). Against this backdrop, we found that extreme cold, extreme heat, and extremely high DTR will significantly drive up CVD mortality in China, with pronounced regional variations. In China, cold seasons may see frequent high-intensity cold periods (*Lei et al., 2022*). Consequently, if winter conditions worsen, cold-related CVD mortality is expected to rise sharply, particularly in southern regions where residents have lower cold adaptability and limited heating resources. Conversely, northern regions are more susceptible to extreme heat. As global average temperatures rise, extreme heat events are likely to become more frequent and intense, placing additional strain on the cardiovascular system and significantly increasing mortality rates (*Wang et al., 2023*). Furthermore, the potential synergistic effect between DTR and extreme heat may exacerbate this health risk, leading to a nationwide increase in CVD mortality (*Lee et al., 2019*). Therefore, addressing the threats posed by extreme temperatures to cardiovascular health necessitates a thorough consideration of geographical and climatic differences. Specifically, southern regions should focus on enhancing heating infrastructure and improving residents' adaptability to extreme cold. Meanwhile, northern regions should prioritize refining heat warning systems and promoting measures to protect residents from extreme heat. Notably, nationwide monitoring of DTR changes, especially under extreme heat conditions, is crucial for timely protective measures to mitigate related health risks. By implementing region-specific and fine-tuned public health interventions and policies, we can more effectively address the health challenges posed by varying climate conditions and mitigate the risks of CVD associated with extreme cold, extreme heat, and extremely high DTR. Additionally, our research findings offer a pivotal reference framework for climate health research and policy-making in countries located at comparable latitudes or sharing comparable developmental statuses. Concurrently, these findings underscore the necessity for nations worldwide to devise and implement public health strategies tailored to their unique climatic features, with the aim of facilitating early warning systems and effectively mitigating the potential hazards posed by extreme climate events on the cardiovascular health of the general populace.

This study has several limitations that warrant consideration. First, as it is difficult to collect all the data on the relationship between temperature and CVD mortality, the data for all cities were not included in the analysis. Second, due to limitations in data collection, the temperature index only considers the impacts of the extreme cold, extreme heat, and

an extremely high DTR. Analysis of other temperature indexes should be included in the future, such as apparent temperature and temperature change between two adjacent days. Third, we did not consider the impact of potential confounding factors. Fourth, this study is a meta-analysis conducted on the foundation of previous research, which inherently suffers from a lack of rigorous differentiation in cardiovascular death data between urban and rural populations. Additionally, the disparities in medical resource allocation and strategies for adapting to extreme temperatures between rural and urban areas pose a limitation to the direct applicability of our findings to both rural and urban regions. Future comprehensive studies on the relationship between different temperature exposures and CVD mortality in China, including heat waves, cold snaps, apparent temperature and TCN, are also needed. These studies should consider more potential variables, such as gender, age, administrative division, education level, availability of heating and cooling, and other socioeconomic conditions.

## CONCLUSIONS

The results of this study indicate that extreme cold, extreme heat and extremely high DTR all lead to an increase in CVD mortality in China, with the impact of extreme cold being the most significant. Residents of northern regions are more susceptible to the effects of high temperatures, while residents of southern regions are more susceptible to the effects of low temperatures. However, under extremely high DTR, there is no significant difference in CVD mortality rates between the southern and northern regions. Therefore, it is crucial to establish a better, more climate-resilient, and environmentally sustainable public health system, such as climate risk integration and real-time climate monitoring, based on the specific climate of different regions to cope with extreme cold and extreme heat, as well as extremely high DTR.

### Funding

This work was funded by the National Natural Science Foundation of China 'Big data-driven Meteorological environment for health risk assessment' [71861026] and Gansu Natural Science Project 'Knowledge-driven Meteorological Environment for Public Health Economic Loss Evaluation' [20JR10RA149]. The funders had no role in study design, data collection and analysis, decision to publish, or preparation of the manuscript.

### Grant Disclosures

The following grant information was disclosed by the authors:
National Natural Science Foundation of China 'Big data-driven Meteorological environment for health risk assessment': 71861026.
Gansu Natural Science Project 'Knowledge-driven Meteorological Environment for Public Health Economic Loss Evaluation': 20JR10RA149.

## Competing Interests

The authors declare there are no competing interests.

## Author Contributions

- Guangyu Zhai conceived and designed the experiments, authored or reviewed drafts of the article, and approved the final draft.
- Ziqing Jiang conceived and designed the experiments, performed the experiments, analyzed the data, prepared figures and/or tables, authored or reviewed drafts of the article, and approved the final draft.
- Wenjuan Zhou analyzed the data, prepared figures and/or tables, and approved the final draft.

## Data Availability

The raw code and data are available in the Supplementary Files.

## Supplemental Information

Supplemental information for this article can be found online at http://dx.doi.org/10.7717/peerj.18355#supplemental-information.

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
