# Peer review of "Differences in cardiovascular disease mortality between northern and southern China under exposure to different temperatures: a systematic review"

_PeerJ, doi:10.7717/peerj.18355_

## Round 0.1 · original submission · Minor Revisions

Both reviewers have provided clear suggestions for you to address. Please be sure to respond to all their comments.

Reviewer 1 ·

Basic reporting

The systematic review and meta-analysis provide a thorough examination of the impact of temperature on cardiovascular disease (CVD) mortality across various regions in China. By stratifying the analysis into extreme cold, extreme heat, and diurnal temperature range (DTR), the study offers valuable insights into regional variations in vulnerability to temperature extremes.

Experimental design

The study adheres to PRISMA guidelines and employs robust statistical methods such as random effects models, sensitivity analysis, and subgroup analysis by region. This approach enhances the reliability and validity of the findings, despite the observed heterogeneity across studies.

Validity of the findings

Findings highlight the urgent need for tailored public health policies to mitigate the adverse effects of temperature extremes on CVD mortality in different regions of China. The identification of differential susceptibilities between northern and southern regions underscores the importance of localized interventions.

Additional comments

How do the findings of this meta-analysis translate to rural areas where access to healthcare resources and adaptation strategies to temperature extremes might differ significantly from urban settings?

Considering the increasing burden of CVD globally, how might climate change influence future trends in CVD mortality associated with temperature extremes in China?

Given that low and middle-income countries bear a disproportionate burden of CVD mortality, how can the findings of this study inform international efforts to address climate-related health disparities beyond China's borders?

Reviewer 2 ·

Basic reporting

Your introduction provides an accurate overview of the context and the public health issue; literature is well referenced and relevant. However, although you reported that exposure to extreme environmental temperatures do contribute to increase in cardiovascular (CV) mortality, how these weather conditions may affect CV health is not clearly specified in the Introduction, but only briefly referred in the Discussion.
Therefore, I suggest to further explicit the pathophysiological linkage between weather conditions (including extreme temperatures), CV health and acute CV diseases in the text, eventually providing some examples.

Experimental design

The paper provides a well-defined research question, with clinical relevance in public health and daily clinical practice. It is clearly stated how the research fills an identified knowledge gap.
A rigorous investigation was performed and methods are described with sufficient detail and information to replicate.

Validity of the findings

All underlying data have been provided; they are robust, statistically sound and controlled.
Limitations of the study are clearly exposed and their implication on results are critically analysed.
Conclusions are well stated, linked to original research question and limited to supporting results.

Additional comments

Your introduction provides an accurate overview of the context and the public health issue; literature is well referenced and relevant. However, although you reported that exposure to extreme environmental temperatures do contribute to increase in cardiovascular (CV) mortality, how these weather conditions may affect CV health is not clearly specified in the Introduction.
I suggest you a recent review by De Vita et al. (J. Clin. Med. 2024, 13, 759. DOI: 10.3390/jcm13030759), which provides an exhaustive compendium about current evidences in the linkage between weather conditions (including extreme temperatures), CV health and acute CV diseases. Therefore, I suggest to further explicit this pathophysiological linkage in the text, eventually providing some examples.

---

## Round 0.2 · accepted · Accept

Dear Authors,

Thank you for your revisions. After carefully reviewing your responses, I am pleased to inform you that you have successfully addressed all of the reviewers' comments. Therefore, the manuscript titled "Differences in cardiovascular disease mortality between northern and southern China under exposure to different temperatures: A systematic review" has been accepted for publication in PeerJ.

Congratulations on your work, and we look forward to seeing your paper published.

Best regards,
Dr. Pedrino

Reviewer 1 ·

Basic reporting

No comment.

Experimental design

No comment.

Validity of the findings

No comment.

Additional comments

No comment.

Reviewer 2 ·

Basic reporting

The pathophysiological linkage between weather conditions (including extreme temperatures), CV health and acute CV diseases are now better explicited in the Introduction and some examples are now available.

Experimental design

No further comments.

Validity of the findings

No further comments.

Additional comments

No further comments.